# Design and analysis of behavioral intervention studies: A Bayesian approach

**Camila Natalia Barragan Ibañez**[1], **Ulrich Lösener**[1], **Nnamdi Moeteke**[2], **Mirjam Moerbeek**[1]*

**1** Department of Methodology and Statistics, Utrecht University, Utrecht, the Netherlands, **2** Department of Community and Public Health, Idaho State University, Pocatello, Idaho, United States of America

* m.moerbeek@uu.nl

## Abstract

To study the effect of a behavioral intervention, it should be compared to a control or an existing treatment in an intervention study. There exist many guidelines in the literature about the design and analysis of intervention studies, including recommendations for a priori sample size determination. The vast majority of these guidelines are based on the framework of null hypothesis significance testing, where a *p*-value is compared to a user-selected type I error rate to determine whether an effect is significant or not. This approach has received severe criticism over the past decades as it has resulted in publication bias, sloppy science, and fraud. The Bayesian approach to hypothesis testing has been developed to overcome some of these drawbacks. The Bayes factor quantifies the relative support in the data for one hypothesis over another hypothesis. The hypotheses do not necessarily have to include a null hypothesis and can be formulated based on observations, findings in the literature, or an expert's opinion. Posterior Model Probabilities, which are a function of the Bayes Factor, can be used to compare a set of hypotheses to one another and select the one most supported by the data. In this paper, we summarize the shortcomings of null hypothesis significance testing, introduce the Bayes factor and Posterior Model Probabilities, explain how they are calculated, and how they are interpreted. We also focus on a priori sample size determination in the Bayesian hypothesis testing framework. We introduce a criterion for sample size determination and a procedure to find the required sample size. We illustrate our methodology using a cluster randomized trial on the effectiveness of an online training in improving primary care doctors' competency in brief tobacco interventions. All analyses are done in R, and we provide the dataset and R syntax for straightforward replication.

**Data availability statement:** Data Availability: The data set corresponds to the study "Effectiveness of online training in improving primary care doctors' competency in brief tobacco interventions: A cluster-randomized controlled trial of WHO modules in Delta State, Nigeria" (https://journals.plos.org/plosone/article?id=10.1371/journal.pone.0292027). Code availability: The data file of the empirical example, all scripts used for analyses and simulations, and simulation results, are available on the Open Science Framework (OSF) at https://osf.io/ejhz6/overview.

**Funding:** The research of UL and CNBI is funded by the Dutch Research Council (file number 406.21.GO.006). More information about the grant can be found in [Bayesian sample size calculation for trials with multilevel data | NWO] (https://www.nwo.nl/en/projects/40621go006). The founder had no role in study design, data collection and analysis, decision to publish, or preparation of the manuscript.

**Competing interests:** The authors have declared that no competing interests exist.

## Introduction

The randomized controlled trial is considered the gold standard for the evaluation of one or more new interventions, therapies, treatments, or pharmaceutical drugs with the aim to change participants' health, attitude, opinion, behavior, and so forth [1,2]. The first randomized controlled trial appears to have been conducted in the 1920s [3] and since then it has seen widespread use in many empirical sciences, including medical and health science, (clinical) psychology, educational sciences, and pedagogics.

The typical randomized controlled trial runs across the lines of a few steps. One first defines the experimental conditions to be compared, selects relevant outcome variables and measurement instruments, and chooses the population and sampling method. Then, an a priori sample size calculation is performed and a sample of sufficient size is drawn from the population. The experimental conditions are then implemented and outcome variables are collected along with demographic and other variables that are supposed to also have an effect on the outcome. One then fits a statistical model to the data to estimate and test differences across experimental conditions on the response variables, adjusted, if necessary, for covariates that were unequally distributed across the experimental conditions. The last step is communication of the results, preferably based on guidelines such as the CONSORT statement [4].

Hypotheses in empirical research are usually tested using the framework of null hypothesis significance testing (NHST), which is also known as the Neyman-Pearson approach to hypothesis testing [5]. The $p$-value quantifies the probability of finding the results in the study or more extreme results, given that the null hypothesis of no differences across experimental conditions is true. If it is smaller than a user-selected significance level, typically set at $\alpha = 0.05$, the null hypothesis is rejected. This means that there is too much evidence against the null hypothesis, and it is concluded that there is a difference across experimental conditions. Despite its widespread use in empirical research, this approach to hypothesis testing has received severe criticism over the past decades and has been considered as the cause of unwanted research practices such as the file-drawer effect, publication bias, sloppy science and fraud, and the replication crisis. In one of the next sections, we will dive deeper into the drawbacks of NHST.

The Bayesian approach to hypothesis testing [6] has been developed as an alternative to NHST. This approach can be used with frequentist (maximum likelihood) estimates and does not require Bayesian *model estimation* per se. It employs the Bayes factor to quantify the evidence in the data for a hypothesis relative to another hypothesis. The Bayes factor is much easier to interpret than the $p$-value and actually provides the information that a researcher is interested in: the relative evidence in the data in support of a specific hypothesis compared to another hypothesis rather than the evidence against the null hypothesis. The hypothesis to be tested does not necessarily have to be the null hypothesis, but can be any hypothesis formulated by the researchers, such as an ordering of group means in a one-way analysis of variance or an ordering of the strengths of various predictors in a multiple regression model.

Such hypotheses are often driven by findings in the literature, expert opinions, or observations in the field, and are also known as informative hypotheses [7]. The Bayes factor can also be used to quantify the relative support in the data for each hypothesis in a set of competing hypotheses.

The Bayes factor was introduced over half a century ago [8] but was not widely used at the time since the marginal likelihood is based on computer-intensive methods [9]. In the last few decades, attention has been paid to further development of the methodology, which has resulted in various R packages to calculate Bayes factors, such as `bain` [10], `BFpack` [11], `BayesFactor` [12] and `baymedr` [13] and the implementation of Bayes factor calculation in the freeware software JASP [14]. In this same period, computers have become much faster. Hence, it is not a surprise that the use of Bayesian methods has increased [15,16].

This tutorial aims to introduce the use of Bayesian hypothesis testing for randomized controlled trials to readers who use empirical data to test hypotheses, such as social and behavioral scientists, and health and medical scientists. The focus will be on the cluster randomized trial (CRT), which is a study design that is often used in these fields [17–21]. Complete clusters of individuals, such as schools, psychotherapy practices, or clinics, are randomly assigned to experimental conditions, and every participant within the same cluster receives the same condition. Although cluster randomization is known to be less efficient than individual randomization, it is often used for practical, administrative, ethical, and financial reasons and the need to avoid contamination [22–24]. Contamination occurs when multiple experimental conditions and a control are available within each cluster, and information about the contents of the experimental condition leaks to participants in the control group. As the data of a CRT have a hierarchical structure, the multilevel (mixed) model [25–27] should be used to avoid an increased risk of a false positive result [28].

We illustrate the Bayesian approach to hypothesis testing on the basis of a recently published paper on the effectiveness of online training in improving primary care doctors' competency in tobacco interventions [29]. We first present a summary of this study, its sample size calculation using NHST, and an interpretation of its results from an NHST perspective using *p*-values. We then give an overview of criticism towards NHST and how the Bayes factor can be used to overcome part of the criticism. Subsequently, we give a non-technical introduction to the Bayes factor and show how it is calculated for our example data. We continue with a priori sample size calculations based on the Bayes factor. We introduce a sample size criterion that is an extension to the criterion used in power analysis for NHST and explain how sample size determination is done on the basis of a simulation study. We then show how the required sample size depends on the cluster size, the number of clusters, the effect size and the intraclass correlation coefficient (ICC, which quantifies the degree of similarity within a cluster). We also calculate the required sample size for the study by [29]. We end this tutorial with a discussion and conclusions, including directions for future research.

We support open science and open software. All analyses in this tutorial are done in the freeware software R, version 4.3.3 [30] and using the package `bain` [10] and our self-written R functions to perform a Bayesian sample size determination. We provide the R syntax in the online supplement so that our analyses can be replicated.

## Motivating example

The dataset used in this paper originates from a previously published cluster randomized trial and is publicly accessible through the original publication.

## Description of the population

The cluster randomized trial [29] aimed to evaluate the effectiveness of online training using WHO modules in enhancing competency in brief tobacco cessation interventions among primary care doctors in Delta State, Nigeria. The study population consisted of primary care doctors employed in government-owned health facilities. There are 25 Local Government Areas (LGAs) that served as clusters and the units of randomization. The inclusion criterion required a minimum of twelve months post-licensure. The exclusion criteria included working in two or more clinics across different LGAs (to prevent

contamination) and being away on in-service training. The three LGAs with fewer than four doctors were excluded based on the calculated necessary minimum size for each cluster. The final analysis included 261 primary care doctors in 22 LGAs (an average of nearly 12 participants per LGA).

## Intervention and control

The intervention was the WHO e-learning course *Training for primary care providers: brief tobacco interventions* [31], which is based on the 5 A's/5R's model for the treatment of tobacco dependence [32]. The 5 A's include 'Ask', 'Advise', 'Assess', 'Assist', and 'Arrange', while the 5R's stand for 'Relevance', 'Risks', 'Rewards', 'Roadblocks', and 'Repeat'. The 5R's are the components of motivational interviewing for patients unwilling to quit at that moment (Fig 1). The WHO course is self-paced and takes an average of six hours to complete. The control group did not receive any intervention until after the study.

## Method of data collection

The study employed a self-administered structured questionnaire adapted from a previously validated iwwwnstrument [33]. Data was collected in two phases (pre-training and six months post-training). Participants received the electronic questionnaire via WhatsApp. It had five sections to assess independent variables (age, years of practice, healthcare level of health facility, etc.), and outcome variables.

## Ethics statement

According to the original study, the researchers obtained ethical approval from the Health Research Ethics Committee of the Delta State University Teaching Hospital in Oghara. The participants signed the written informed consent forms provided by the researchers. The details can be found in the original study [29].

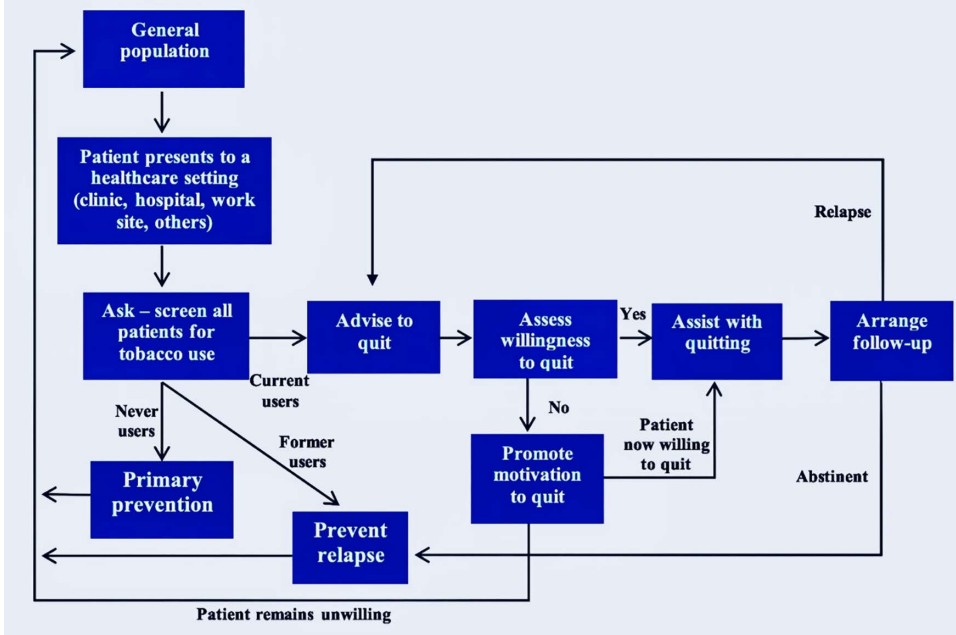

**Fig 1. The 5As/5Rs model for treatment of tobacco use and dependence in primary care.** Reproduced from [29].

Considering that the present study involves the secondary use of pseudonymised data containing personal information, ethical approval was requested from Utrecht University. The Ethics Review Board of the Faculty of Social and Behavioral Sciences granted approval for the use of the data (Approval number: 25–0068).

## Outcome variables

The study had four outcome variables with respect to brief tobacco cessation interventions using the 5 A's/5R's model: Knowledge, Attitude, Confidence, and Practice. The change in the study's confidence variable from baseline to post-training will be used to illustrate the methodology of Bayesian hypothesis and sample size determination in this paper. The confidence variable was measured with 8 questions that required participants to rate their self-efficacy (in performing the various activities of the 5 A's/5R's model) on a 3-point scale: "not at all confident", "a little confident" and "fully confident"). A score of zero was awarded for "not confident", one for "a little confident" and two for "confident". Total confidence scores could range from 0 to 16, and we assumed this outcome variable to be approximately continuous. Note that in cases of more uncertainty about scale properties, ordinal alternatives (such as generalized linear models [34]) should be used.

## Sample size determination

The minimum sample size per group for a CRT, $N_C = N_I * VIF$, where $N_I$ is the minimum sample size for an individual randomized controlled trial, and VIF is the Variance Inflation Factor [35], $VIF = 1 + (m-1)\rho$, where m is the average cluster size, and $\rho$ is the ICC [36]. The minimum sample size per study arm for an individual randomized controlled trial ($N_I$) is calculated using the following formula [37].

$$N_I = \frac{2 * (z\alpha/2 \; + z_\beta)^2 * S^2}{\delta^2}$$

(1)

Where S is the pooled standard deviation of both comparison groups, δ is the anticipated difference in the two group means ($\mu_1 - \mu_2$); $Z_{\alpha/2} = Z_{0.05/2} = 1.96$ at a type-I error rate of 0.05; $Z_\beta = Z_{0.8} = 0.842$ at a power of β = 0.8. A previous educational intervention study on tobacco cessation counselling among health professionals obtained an effect size of ($\mu_1 - \mu_2$)/S = 0.52 [38]. Therefore, $N_I = 15.7/0.522^2 \approx 58$ per arm.

The ICC for tobacco dependence treatment at the primary care practice level had been estimated to be 0.054 [39]. As such, the VIF for this study was VIF = 1 + (12.56–1)×0.054 = 1.624, and the minimum sample size per arm for the CRT, $N_C = N_I \times VIF = 58 \times 1.624 \approx 94$. To take care of possible attrition this was increased by 10% to 103 per arm (206 in total).

## Results

For this paper, the free software R and R-studio with the packages *lmer* (version 1.1−37) [40] and *lmerTest* (version 3.1−3) were used for data analysis. To allow for clustering, the test of significance for difference in change in confidence between the control and intervention groups was done using a multilevel mixed model, with study group modelled as a fixed effect and LGA of practice entered as a random effect. Note that, alternatively, an ANCOVA can be employed in situations with two measurements. However, in order to preserve comparability with the original study by Moeteke et al. [29] we likewise use a multilevel regression model. The model was also adjusted for age, years of practice, and healthcare level of health facility. The ICC was 0.042, which indicates that the observations within clusters were only slightly more similar than those from different clusters, i.e., only 4.2% of the variance was located at the level of the LGA.

The difference in mean of change in confidence scores between the study groups was 1.934 (SE = 0.555; t = 3.493; p = 0.006), with an effect size of 0.590 according to Hedges and Hedberg [41] and Hedges [42] (Table 1), and it was highly significant at a type-I error rate of α = 0.05. An effect size of 0.59 is a medium to large treatment effect according to Cohen's d for two independent groups. The result from this analysis is slightly different from that obtained in the original

**Table 1. Model estimates of the effect of study group on change in total confidence scores, adjusted for age, years of practice, and healthcare level of health facility.**

**Random effect**

|  | Variance | Standard deviation |  |
|---|---|---|---|
| LGA* of practice | 0.453 | 0.673 |  |
| Residual | 10.277 | 3.206 |  |

**Fixed effects**

|  | Estimate | Standard error | df* | t value | p-value | Effect size |
|---|---|---|---|---|---|---|
| Intercept | −3.724 | 2.048 | 163.296 | −1.818 | 0.071 |  |
| Study group | 1.934 | 0.555 | 9.470 | 3.483 | 0.006 | 0.590 |
| Age | 0.081 | 0.054 | 252.406 | 1.503 | 0.134 |  |
| Years of practice | −0.079 | 0.054 | 252.087 | −1.456 | 0.147 |  |
| Factor (HLoHF) 2 | 0.156 | 0.690 | 253.780 | 0.226 | 0.821 |  |
| Factor (HLoHF) 3 | 0.953 | 0.773 | 106.267 | 1.233 | 0.220 |  |

Only estimates for secondary and tertiary levels of healthcare are included in the table with fixed effects. Primary level serves as reference. Degrees of freedom are adjusted according to Satterthwaite's method.

Abbreviations: "HLoHF" is the Healthcare Level of Health Facility, "df" indicates the degrees of freedom, "LGA" is the Local Government Area

study [29] most likely because a typographical error in the dataset has been corrected. However, in both analyses, the effect of the study group on change in confidence is highly significant.

## Shortcomings in null hypothesis significance testing

Null hypothesis significance testing (NHST) has long been the prevailing inferential method in behavioral and medical research, even after receiving severe criticism in the past decades [5,6,43–47]. In this paper, we briefly discuss the main theoretical and practical shortcomings of the NHST approach and introduce Bayesian hypothesis evaluation as an alternative.

### Misinterpretations of *p*-values and confidence intervals

In most studies, researchers strive to gain insight into the veracity of their hypotheses given the data they collected. However, this cannot be achieved within the framework of NHST [48,49]. The main product of a statistical test in a NHST analysis is the *p*-value, which is the probability of obtaining at least the observed difference or effect given that the null hypothesis is true. This probability, however, is the *inverse* of what most researchers are interested in, which is the probability of their hypothesis being true given the data. As can be shown in simulation studies, the *p*-value gives little insight into the latter probability [50] and thus answers a question that researchers are not typically asking [51]. Nevertheless, many researchers misinterpret the *p*-value as the probability of the null being true given the data (inverse fallacy) resulting in inferential errors [52,53]. In the Bayesian framework, on the other hand, the probability of a hypothesis being true given the data *can* be obtained, given that the set of hypotheses under consideration includes the true hypothesis [48,54].

The inverse fallacy is not the only deeply established misconception about the *p*-value. Another common one is mistaking the *p*-value as type-I error (false positive) probability. The probability of committing a type-I error in the long run is denoted as α and typically fixed at 0.05 [55]. Contrary to the *p*-value, α is fixed before any data is analyzed and thus does not depend on the data. Regrettably, many researchers and scholars misunderstand the *p*-value as a "data-adjusted" type-I error rate, and to make matters worse, this misconception is printed in various textbooks in disciplines like psychology, health research, behavioral sciences, biology, and econometrics [56].

Another common misinterpretation related to estimation rather than hypothesis testing pertains to confidence intervals (CIs) which quantify the uncertainty around a point estimate. The correct interpretation of a 95% CI is that if the experiment is repeated and the CI recalculated many times, 95% of these CIs contain the true population value, assuming unbiased estimation and random sampling from the same population. This rather cryptic interpretation is frequently misunderstood as the probability of the true value being within the CI being equal to 0.95 [57]. The latter interpretation is valid only for *credible intervals*, the Bayesian counterpart of CIs [58,59].

### Arbitrariness of the α-level

The establishment of cut-off points for levels of significance can be attributed to Fisher's work [60], which was mainly concerned with applying his statistical theory to biology and agronomy. He mainly used 0.05 and 0.01 as α-levels without the ambition of asserting interdisciplinary conventions. Nonetheless, these values became firmly established in social and behavioral sciences. This was convenient at a time when computers were rare, and tables provided overviews of critical values for these α-levels for popular distributions [48]. Today, however, the persisting mindless reliance on this arbitrary dichotomous decision rule where "$p < α$ is significant and $p > α$ is not" harms scientific progress [47]. As stated in [61]: "Surely, God loves the .06 nearly as much as the .05". The exact *p*-value as a measure of evidence against the null is not even of interest to the researcher since its only purpose is to be compared against α in order to make a dichotomous decision (which Fisher surely would have disagreed with; [48]). This decision, however, does not inform as to the degree to which one believes in the veracity of the hypothesis of interest and thus does not provide a satisfactory answer to the typical research question at hand [62,63]. In the Bayesian framework of hypothesis testing, similar thresholds have been proposed [8,54] but they serve as interpretation guidelines rather than dichotomous decision rules. Nonetheless, we advocate for letting the evidence speak for itself instead of relying on cutoffs. Note, however, that for the purpose of sample size determination, establishing some threshold value is obligatory (see later sections).

### Sample size

The relationship between significance and sample size is another obscure and often misunderstood aspect of NHST. As the sample size increases, statistical power increases, and small population effects are recognized more easily [64]. This means that with a large enough sample, even minuscule, practically irrelevant effects become statistically significant [54]. It is also worth mentioning that sample size has no implication about the quality of the data. It is thus erroneous to assume that statistical significance implies real-world importance and to interpret a *p*-value without careful consideration of its accompanying effect size [48,65].

The belief that a significant result from a study with a larger sample is more reliable than that from a study with a smaller sample is another fallacy stemming from a misconception about type-II errors (false negative). With larger samples, the chance of committing a type-II error decreases, but the long-run chance of committing a type-I error stays fixed at α [48]. Thus, a significant result has the same probability of being spurious regardless of sample size.

### Summary

It is important to point out that no method of inference that we know of today will result in flawless application in empirical science. Our aim is not to condemn NHST via *p*-values per se, considering that *p*-values, when constructed and interpreted correctly, behave exactly as they should [66]. In some situations (when there is no prior information available, when controlling long-run Type-I error probabilities is of importance), NHST can even be more suitable than Bayesian inference [67]. Rather, the problem lies in the highly prevalent misinterpretation of *p*-values, which are often characterized by a rigid reliance on the $p < .05$ criterion [68] and the neglect of accompanying information such as effect sizes and confidence intervals [69,70]. While Bayesian hypothesis evaluation is not immune to misinterpretations and misuse [71–74], we believe that it addresses some of the major flaws in the application of NHST and should be an accessible alternative for applied researchers.

## Bayesian hypothesis evaluation

Bayesian hypothesis evaluation via the Bayes Factor (*BF*) is a viable alternative to NHST which does not suffer from some of the aforementioned shortcomings [54,75]. Note that this method can be applied to estimates of model parameters obtained from frequentist models and does not require Bayesian estimation. In the Bayesian framework of hypothesis testing, the prior odds $\frac{P(H_0)}{P(H_1)}$ reflect a priori expectations about the hypotheses and the posterior odds $\frac{P(H_0|D)}{P(H_1|D)}$ reflect the a posteriori belief about the hypotheses after taking into account the data at hand (D). The *BF* marks the shift from prior to posterior odds, that is, how a belief about the relative veracity of two hypotheses is updated when considering data [76].

$$\frac{P(H_0|D)}{P(H_1|D)} = BF_{01} * \frac{P(H_0)}{P(H_1)}$$

(2)

For the sake of this tutorial paper, we will assume equal prior probabilities for each hypothesis. The BF thus quantifies the relative evidence in the data for a pair of competing hypotheses and has the following straightforward interpretation [6]. If $BF_{01} = 2$ then there is twice as much evidence for $H_0$ compared to $H_1$, and conversely, if $BF_{01} = 0.5$ then there is twice as much evidence for $H_1$ compared to $H_0$. If $BF_{01} = 1$, then there is equal support for $H_0$ and $H_1$. Note that, contrary to NHST, it is actually possible to provide evidence *for* the $H_0$, though the use of a null hypothesis is not compulsory [77]. The following heuristic guidelines for interpreting the strength of a BF have emerged in the literature: Sir Harold Jeffreys [8] as well as Kass and Raftery [54] consider $BF > 3.2$ to be "substantial evidence", whereas Lee and Wagenmakers [78] more conservatively consider $3 < BF < 10$ "moderate" and $10 < BF < 30$ "strong evidence". All of the above authors label $1 < BF < 3$ either "anecdotal" or "barely worth mentioning". Note that, as stated in the previous section, our recommendation is to abstain from referring to threshold values and let the reader decide upon the meaningfulness of a BF, in line with Hoijtink et al. [6].

While a number of different ways to calculate the BF can be found in the literature, we will employ the Approximated Adjusted Fractional Bayes Factor (henceforth: BF) as it stands out for its simplicity [79]. This way of approach to calculating the BF has the advantage that it does not require the researcher to specify a prior distribution for the parameter(s) in question as it uses a default fractional prior that is constructed from a part of the data. Also, its computation does not rely to Markov Chain Monte Carlo sampling mechanisms and is therefore much faster, a property that is especially desirable in simulation settings. This BF is implemented in the open-access R packages *bain* [10] and *BFpack* [80]. Other ways of computing the BF are implemented in software packages such as *BayesFactor* [81] and *brms* [82]. Rather than elaborating on the mathematical details of the calculation of the BF (for a detailed overview see [79]), we illustrate its use at the hand of example hypotheses about a regression parameter. The regression parameter of interest is $\beta_{StudyGroup}$, representing the difference in the outcome between the control group and the intervention group. The hypotheses under consideration are $H_0: \beta_{StudyGroup} = 0$ (no difference between groups), $H_1: \beta_{StudyGroup} > 0$ (the intervention group does better than the control group), $H_c:$ neither $H_0$ nor $H_1$ thus: $H_c: \beta_{StudyGroup} < 0$, and $H_u: \beta_{StudyGroup}$, where $H_u$ is the so-called unconstrained hypothesis which puts no constraints on the parameter at all. The unconstrained hypothesis states that "anything could be going on" and can be looked upon as the counterpart to the frequentist null hypothesis which states that "nothing is going on". The BF of $H_1$ against the unconstrained is defined as the ratio of its fit and complexity.

$$BF_{1u} = \frac{fit_1}{complexity_1}$$

(3)

The complexity of a hypothesis indicates the (absence of) parsimony and is operationalized as the agreement of the prior distribution and the parameter space under the hypothesis. Generally, the prior distribution represents the researchers' a priori belief about a model parameter. However, when using the Approximate Adjusted Fractional Bayes Factor, we use a so-called default prior which is automatically constructed from a fraction of the data [83], removing a potentially

burdensome task for the researcher. The exact fraction is determined by the parameter $b$, typically fixed at $b = \frac{1}{N_{eff}}$ where $N_{eff}$ is the effective sample size. The effective sample size reflects the amount of information in the data corrected for its nested structure (i.e., subjects in clusters in a cluster randomized trial) and is defined as $N_{eff} = \frac{n_1 n_2}{1+(n_1-1)ICC}$, where $n_1$ is the cluster size and $n_2$ is the number of clusters [84]. Using the inverse of the effective sample size minimizes the amount of data used for the prior in line with the logic of a minimal training sample [79]. For a more robust prior, $b$ can be set to $b = \frac{2}{N_{eff}}$ or to $b = \frac{3}{N_{eff}}$. This default prior is normally distributed with its center around the boundary of the two hypotheses under investigation; in the case of $H_1$ vs. $H_u$, the boundary is zero [6]. The fit, on the other hand, indicates how well a hypothesis describes the data and is defined as the agreement of the posterior distribution and the parameter space under the hypothesis. The posterior distribution represents one's belief about the model parameter after combining a priori expectations (the prior) and the data. The normally approximated posterior distribution is centered around the maximum-likelihood estimate and its standard deviation is equal to the standard error of the estimate [79]. For an illustrative example of the calculation of the BF along with a visualization of the fit and complexity of both an equality and inequality constrained hypothesis, see S1 File and S2 Fig. Another way to compare numerous hypotheses is using the Posterior Model Probabilities (PMPs), which represent the relative degree of support for each hypothesis in the set given the data. Assuming that all hypotheses are equally likely, the PMPs can be computed as [85].

$$PMP = \frac{BF_{iu}}{\sum_i BF_{iu}}$$

(4)

The Bayes factors are, then, transformed to a scale between 0 and 1, allowing the comparison in a set with more than two hypotheses.

## Bayesian results for the motivating example

In the following, we illustrate this method using the data from the motivating example [29], which we accessed between October 2024 and January 2025, and the R package `bain` (version 0.2.8; [10]). The vignette for this package can be found online and offers an extensive tutorial with various examples. For a step-by-step tutorial for the Bayesian analysis in this paper, see the Rmarkdown file in the supplementary material available online (S3 File). The results are displayed in Table 2. Four hypotheses are being evaluated against each other, among which are our research hypotheses $H_0$ and $H_1$ as well as the unconstrained hypothesis $H_u$ which poses no constraints on the model parameter and the complement hypothesis $H_c$ which covers the entire parameter space left uncovered by either $H_0$ or $H_1$. These latter two hypotheses can be looked upon as safeguards, which would be supported if the set of research hypotheses fits the data poorly [6]. The second and third column of Table 2 display the relative fit and complexity for each hypothesis (except for the unconstrained, as its fit and complexity are always maximal). The third column shows the BF against the unconstrained, which is simply the ratio of fit to complexity of each hypothesis. From this column, we can gather that the support in the data

**Table 2. Results of the Bayesian hypothesis evaluation.**

| Hypothesis | Fit | Complexity | $BF_u$ | $BF_c$ | $PMP_a$ | $PMP_b$ | $PMP_c$ |
|---|---|---|---|---|---|---|---|
| $H_0$: $\beta_{StudyGroup} = 0$ | 0.002 | 0.053 | 0.031 | 0.031 | 0.015 | 0.01 | 0.015 |
| $H_1$: $\beta_{StudyGroup} > 0$ | 1 | 0.5 | 2 | 4032.744 | 0.985 | 0.66 | 0.984 |
| $H_u$: $\beta_{StudyGroup}$ | – | – | – | – | – | 0.33 | – |
| $H_c$: neither $H_0$ nor $H_1$ | 0 | 0.5 | 0 | – | – | – | 0 |

$BF_u$, Bayes Factor versus the unconstrained hypothesis $H_u$; $BF_c$, Bayes Factor versus the complement hypothesis $H_c$; $PMP_a$, posterior model probabilities between the hypotheses under consideration $H_0$ and $H_c$; $PMP_b$, posterior model probabilities between $H_0$, $H_1$ and $H_u$; $PMP_c$, posterior model probabilities between $H_0$, $H_1$ and $H_c$.

for $H_0$ is less than the support for the unconstrained, more specifically, the support in the data for the unconstrained is $1/0.031 \approx 32$ times more than that for $H_0$. The support for $H_1$ is twice as much as the support for the unconstrained. Note that in the case of one parameter, for inequality ($<,>$) constrained hypotheses such as $H_1$ the $BF_{1u}$ is bound at 2 as the complexity of $H_1$ will always be 0.5 and its fit cannot exceed 1. In the next column, we see the respective BF against the complement which, other than $BF_u$, does not have an upper bound. We see that the data supports $H_1$ roughly 4232 times more than the complement. Note that for equality constrained (=) hypotheses with one parameter, the complement is equal to the unconstrained. The last three columns of Table 2 show the PMPs. Note that the PMPs within a column will always sum up to 1. Here, the set of hypotheses under consideration is relevant. In the case of $PMP_a$, only the two research hypotheses $H_0$ and $H_1$ are considered. For $PMP_b$, the unconstrained hypothesis is added to the set of hypotheses under consideration and for $PMP_c$, the research hypotheses are compared against the complement. While the researcher may choose which version of the PMP to report, we recommend using $PMP_c$ as the inclusion of the complement guarantees that the whole parameter space is covered while avoiding overlap with the other hypotheses.

The data clearly supports $H_1$ the most out of all candidate hypotheses, as the PMP is the largest for $H_1$ in each set. In the case of two competing research hypotheses, it is sensible to report the BF of these competing hypotheses, i.e., $BF_{01}$ or $BF_{10}$ in order to quantify the relative evidence for the hypothesis of interest. The $BF_{10}$ can be easily calculated as the ratio of the respective $BF_u$, that is $BF_{10} = \frac{BF_{1u}}{BF_{0u}} = \frac{2}{0.031} \approx 64.52$. This means that the data supports $H_1$ about 65 times more than $H_0$. From all this, we conclude that $H_1$ is by far the best hypothesis in the candidate set. Note that the PMPs cannot per se detect the "truth" as the true hypothesis may not be part of the set of hypotheses under consideration. If all hypotheses in the set poorly represent the true data-generating process, then the PMPs will only identify the least bad hypothesis [6].

## A priori Bayesian sample size determination

### Bayesian power

Bayesian power is defined as the probability ($\eta$) that the Bayes factor $BF_{ii'}$ for testing hypotheses $H_i$ versus $H_{i'}$ exceeds a user-specified threshold ($BF_{thres}$) when hypothesis $H_i$ is true. For hypothesis set 1, we determine sample size such that for both hypotheses a sufficient power is achieved,

$$P\left(BF_{ii'} \geq BF_{thres} | H_i\right) \geq \eta \ \text{ and } P\left(BF_{i'i} \geq BF_{thres} | H_{i'}\right) \geq \eta \tag{5}$$

It is important to note that it is also possible to use different thresholds and probabilities for the two hypotheses in the set, however in this paper we restrict to equal values. Hypothesis set 1 is used for comparing a hypothesis stating a zero-treatment effect ($H_0: \beta_{StudyGroup} = 0$), and an alternative hypothesis stating that the treatment effect is greater than zero ($H_1: \beta_{StudyGroup} > 0$).

For hypothesis set 2, the sample size is determined such that the power criterion is met for one of the hypothesis in the set

$$P\left(BF_{ii'} \geq BF_{thres} | H_i\right) \geq \eta \tag{6}$$

Hypothesis set 2 compares the pair of hypotheses $H_1: \beta_{StudyGroup} > 0$ versus $H_2: \beta_{StudyGroup} \leq 0$. For this set the power criterion must be satisfied for only one of the hypotheses, because the one hypothesis is the complement of the other.

The threshold ($BF_{thres}$) and the probability ($\eta$) can be set to different values according to the desired degree of evidence and the potential consequences of incorrectly finding more support for the incorrect hypothesis, respectively. One potential criterion for the selection of values of these parameters is the level of importance attached to the research. In the context of high-stakes research, where compelling evidence is required, the threshold and probability are set at higher values. Conversely, in low-stakes studies, the researcher may opt for lower values. Another potential criterion for the selection

of the parameters is the objective of the study. For instance, when the objective of the research is to replicate a well-established phenomenon, the values of probability and threshold may be increased. On the other hand, if the objective is exploratory, the values can be set to a lower level. An additional criterion is the cost of carrying out the research. In studies requiring a considerable amount of resources, researchers may aim to ensure a sufficient probability of finding the desired degree of evidence, consequently the probability ($\eta$) may be set to high values.

Fig 2 illustrates the concept of Bayesian power. The left panel shows the distribution of the Bayes factor $BF_{01}$ under the null hypothesis $H_0$, whereas the right panel represents the distribution of the Bayes factor $BF_{10}$ when the inequality-constrained hypothesis $H_1$ is true. The plots were built using the same values that were used in the motivating example for planning, an ICC of 0.054, an effect size of 0.52, and 23 clusters. In the left panel the cluster size was 60 participants per cluster, whilst in the right panel a cluster size of 6 was used to achieve the same level of power. In both panels, the grey area indicates the proportion of Bayes factors that exceed $BF_{thres} = 5$, with the proportion in both panels being at least $\eta = 0.8$. It is suggested that the researcher selects the highest sample size to ensure that the aforementioned criterion is satisfied. In this example that would correspond to 60 individuals per cluster.

### Required ingredients for Bayesian sample size determination

The required sample size in hypothesis testing is influenced by various elements that researchers must carefully consider. For a CRT, these elements include the effect size, ICC, and the number of clusters or cluster sizes. In addition to these ingredients, when employing the Bayes factor in hypothesis testing, researchers must also decide on the probability ($\eta$), the Bayes factor threshold ($BF_{thres}$), and the pair of hypotheses $H_i$ and $H_{i'}$ to be evaluated.

As shown in the literature [21,86], both the effect size and the ICC impact the required sample size in the context of NHST. The sample size is contingent upon the magnitude of the effect size: small effect sizes require a larger sample size, whereas larger effect sizes reduce the required sample size. In contrast, the ICC exhibits an inverse relationship with the sample size. A higher ICC indicates a larger degree of similarity between outcomes of individuals in the same cluster and, thus, the effective sample size decreases. As a result, more clusters or more individuals per cluster are required to obtain

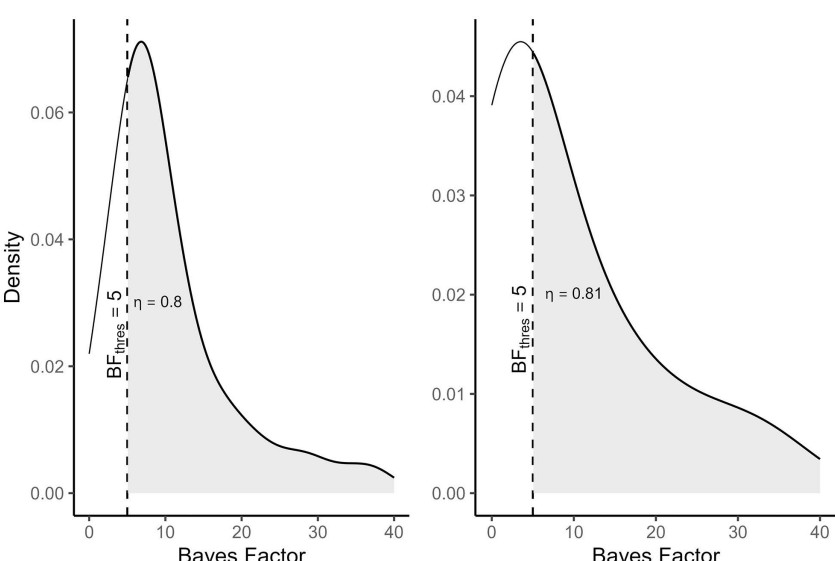

**Fig 2. The distribution of the Bayes factors for the null hypothesis (left) and alternative hypothesis (right).** The dashed line represents the threshold for the Bayes factor, set at 5 in this example. The grey area represents a probability of at least 0.80, which surpasses the desired probability of $\eta = 0.8$.

the same power level as in a trial without nesting of individuals in clusters [87]. The selection of the effect size and ICC may be based on previous studies or following experts' advice. Suggested values for the ICC can be found in the literature (e.g., [88,89]). Additionally, a compilation of reported ICCs across different research fields is also available [86].

Another critical consideration in sample size determination is the trade-off between the number of clusters and the cluster sizes in the NHST. Small number of clusters require large cluster sizes and small cluster sizes require large number of clusters to reach the same power level. However, the increase in power due to increasing the cluster size is limited when the number of clusters is too low, hence a very large cluster size cannot always guarantee a sufficient power level [35]. Taking this into consideration, researchers must fix one of the sample sizes to determine the other one in the proposed methodology. The cluster size can be set at the average number of participants per cluster. The number of clusters can be set at the number of clusters that the researcher aims to recruit for the study. In our methodology for sample size determination, we assume equal number of clusters per treatment condition, as well as equal cluster sizes, which is conventional in the optimal design literature. Even though these assumptions might seem strict, [90] proved that equal allocation can be optimal when costs and variances do not vary across treatment conditions. In addition, [91] have demonstrated that the loss of efficiency due to variation in cluster sizes rarely exceeds 10% for NHST approach. Thus, if there is a consideration of unequal cluster sizes, 11% more clusters are needed to compensate.

As mentioned in the previous section, the probability and the Bayes factor threshold are crucial elements that can be selected based on the objective of the study. High-stakes studies would call for compelling evidence and thus require higher values for probability and threshold. One example of such a study is the development of a treatment for anorexia nervosa based on the use of metreleptin, which is an analogue of the key hormone in homeostasis of body weight [92]. Given that it is fundamental to determine the safety and effectiveness of this new treatment for a disorder that can be life threatening, researchers may opt for high thresholds and probabilities, such as $BF_{thres} = 10$ and $\eta = 0.9$, in order to collect evidence of clear benefit over the control condition. In contrast, in studies with less severe consequences, researchers may adopt a lower threshold and probabilities. An example for which high thresholds and probabilities are not required is a study testing different interventions to reduce anxiety in athletes before a competition. Considering the repercussions of not using the best intervention to manage anxiety prior a competition are not severe, researchers may opt to use low thresholds and probabilities, such as $BF_{thres} = 3$ and $\eta = 0.8$.

The choice of a hypothesis set depends on the research question and the comparisons being conducted. Researchers may be interested in using Hypothesis set 1 when evaluating the effectiveness of a particular treatment in comparison to the control condition. For instance, consider a study where the effect of virtual reality exposure therapy (VRET) to treat social anxiety disorder is evaluated. The null hypothesis states that there is no difference in the treatment effect of VRET and the control group, while $H_1$ states that the treatment effect of the therapy is larger than zero. On the other hand, Hypothesis set 2 may be of interest to researchers when the aim is to compare two treatments or interventions, with one of the treatments serving as the point of reference and the test being whether the new treatment has a better or worse effect than the reference treatment. An example is a study to compare the effect of two treatments for social anxiety disorder: cognitive behavioral therapy (CBT) accompanied by medication and CBT accompanied by VRET. The first hypothesis ($H_1$) postulates that the effect of CBT coupled with VRET is larger than the effect of CBT with medication. In comparison, $H_2$ indicates that the conventional treatment with CBT and medication has a better or similar effect in reducing social anxiety disorder symptoms than CBT with VRET.

In the GitHub repository, researchers will find the necessary scripts to determine the sample size. The functions `SSD_crt_null` and `SSD_crt_inform` determine sample sizes for hypothesis sets 1 and 2, respectively. Both functions determine either the cluster size for a user-specified fixed number of clusters, or the number of clusters for a user-specified fixed cluster size. It is important to note that these functions set fraction b to the minimal training sample approach meaning that $b = \frac{1}{N_{eff}}$ by default [79,83]. It is possible, however, to set fraction b to another value. Given the impact of fraction b on the evaluation of the null hypothesis, researchers are advised to carry out a sensitivity

analysis in order to assess the stability of the sample size. For further details about the effect of the different elements in the sample size determination, the reader is referred to [93] and the accompanying Shiny app (utrecht-university .shinyapps.io/BayesSamplSizeDet-CRT/) [94].

## Application

A post hoc power analysis was performed using the observed effect size (0.59) and ICC (0.042), as well as the sample sizes from the motivating example [29]. As thresholds $BF_{thres}$ of 3, 5, and 7 were considered, with the probability of exceeding the threshold set at $\eta = 0.8$. The results showed that for a threshold of 3, the power criterion was not met for the null hypothesis ($P(BF_{01} \geq 3|H_0) = 0.844$), yet it was met for the alternative hypothesis ($P(BF_{10} \geq 3|H_1) = 0.992$). A similar outcome is observed when the threshold is set at 5, with the power criterion for the null hypothesis not being met ($P(BF_{10} \geq 5|H_0) = 0.655$), while the criterion for the alternative hypothesis was met ($P(BF_{10} \geq 5|H_1) = 0.990$). Furthermore, the power criterion was not met for both hypotheses when the Bayes factor is increased to 7: ($P(BF_{01} \geq 7|H_0) = 0.481$ and $P(BF_{10} \geq 7|H_1) = 0.990$).

The a priori sample size determination was carried out with an ICC of 0.054 and an effect size of 0.52, in accordance with the prior sample size determination presented in the motivating example [29]. As in the motivating example, only hypothesis set 1 was considered. The threshold was set to 3, and the probability was set to 0.8. In the case of determining the number of clusters the cluster size was fixed at 12, which was the minimum cluster size in the motivating example. While, for determining the cluster size, the number of clusters was fixed at 23. The results indicate that when the cluster size is fixed at 12 doctors, at least 16 clusters must be recruited to ensure that the power criteria are met: ($P(BF_{01} \geq 3|H_0) = 0.809$ and $P(BF_{10} \geq 3|H_1) = 0.903$). On the other hand, when the total number of clusters is 23, at least 10 doctors per cluster must be recruited in order to meet the power criteria: ($P(BF_{01} \geq 3|H_0) = 0.831$ and $P(BF_{10} \geq 3|H_1) = 0.942$).

A simulation study was conducted to determine the a priori sample size using common values that can be found in CRTs as well as values obtained from the motivating example. The factors and their levels are the following:

- The cluster size is fixed at 12 when the number of clusters is determined. This is the same value as the average cluster size reported in the empirical example.

- The number of clusters is fixed at 23 when the cluster size is determined. This is the number of clusters in the empirical example.

- The set of hypotheses to compare are the aforementioned Hypothesis set 1 and Hypothesis set 2.

- The effect size. The levels considered are Cohen's $d = 0$, 0.2, **0.4**, 0.59, 0.7, and 0.8. These represent no effect, small, medium, and large effects, as well as the effect of 0.6 found in this study.

- The intraclass correlation coefficient. The levels considered are 0.01, **0.025**, 0.042, 0.075, 0.1. These levels represent small, medium, and large correlation intraclass correlation coefficients, and include the intraclass correlation coefficient of 0.042 found in the data.

- The Bayes factor thresholds ($BF_{thres}$) are 1, **3**, 5, and 7.

- The probabilities ($\eta$) of finding Bayes factors larger than the Bayes factor threshold are 0.7, **0.8**, and 0.9.

Each panel in Figs 3 and 4 illustrates the effect of one of the latter four factors (effect size, ICC, threshold, and probability) on the required sample size, keeping the values of the other three factors constant at the boldfaced value given above. Fig 3 shows the minimum cluster size for which the power criterion is met as a function of each of these four factors separately. Fig 4 shows the minimum number of clusters per treatment condition for which the power criterion is met for each of these factors separately.

The results for effect size and ICC are consistent with the existing literature on sample size in NHST. The top left panels in [Figs 3] and [4] show that small effect sizes require larger sample sizes, while large effect sizes need smaller sample sizes, particularly for hypothesis set 1. The top right panels in both figures demonstrate that, with regard to hypothesis set 1, an increase in the ICC is associated with an increase in the required sample size. In contrast, the sample size remains constant as a function of the ICC for hypothesis set 2. This indicates that the chosen minimum sample size is sufficient to meet the power criteria across various ICC values, this minimum is 6 individuals in [Fig 3] and 8 clusters in [Fig 4].

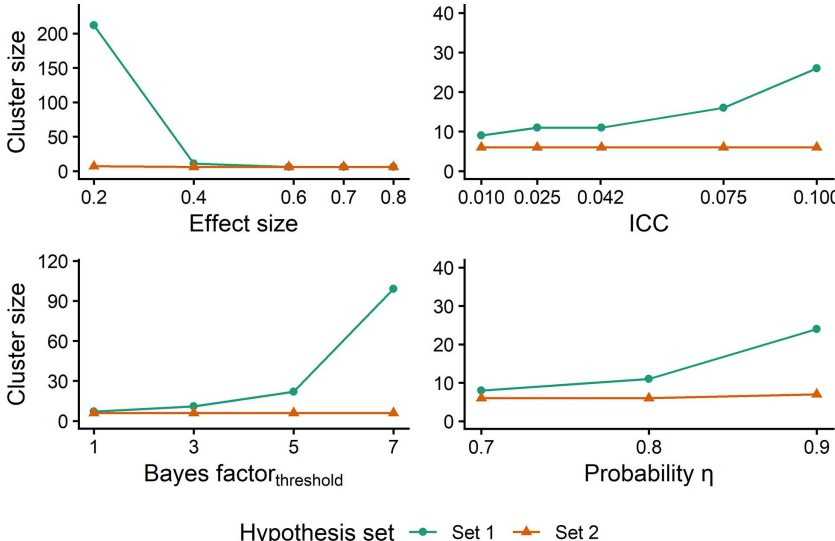

**Fig 3. Required cluster size as a function of effect size, ICC, Bayes factor threshold and probability.** ICC stands for intraclass correlation coefficient. The minimum cluster size chosen in the simulation is 6 individuals per cluster.

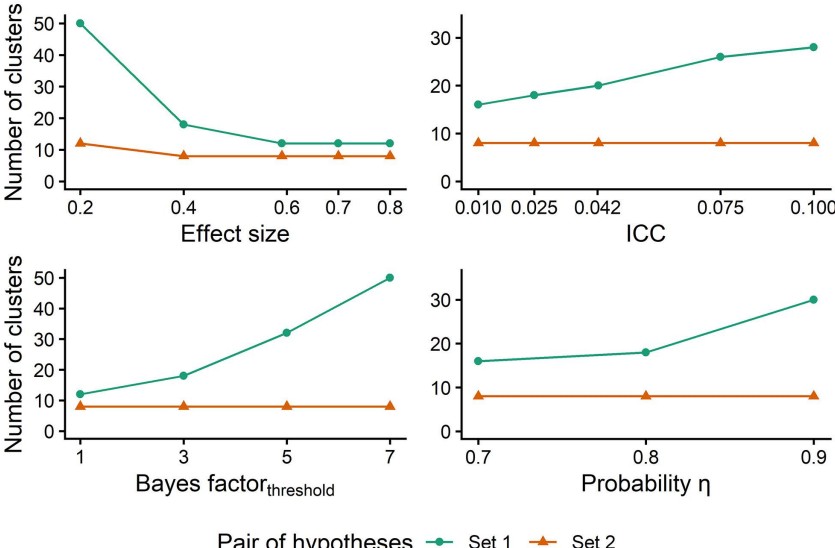

**Fig 4. Required number of clusters as a function of effect size, ICC, Bayes factor threshold and probability.** ICC stands for intraclass correlation coefficient. The minimum number of clusters chosen in the algorithm is 8.

The bottom left panels of Figs 3 and 4 illustrate the number of clusters as a function of the threshold for hypothesis set 1 and 2. It noticed that an increase of the threshold results in an increase of the sample size for hypothesis set 1. In contrast, the sample size is constant across the different thresholds for hypothesis set 2. When the cluster size is fixed, most cases reach the power criterion with 8 clusters in total (4 in treatment and 4 in control), which is the minimum number of clusters that we used in the simulation study. So with a small number of clusters the Bayesian power criterion is achieved, irrespective of the value of the threshold. Had we used an even smaller number of clusters in our simulation, then perhaps fewer than 8 clusters would have sufficed for the small values of the threshold. However, with very small number of clusters the results of the study are not generalizable. As is shown in Fig 3, when the number of clusters is fixed, most cases reach the power criterion with 6 individuals per cluster for hypothesis set 2. For hypothesis set 1, larger thresholds result in larger cluster sizes.

The bottom right panels in both figures show a similar effect of the change in probability (η) on the sample sizes. For larger probabilities, the required sample size is larger for hypothesis set 1, while for hypothesis set 2 the sample size is constant even after increasing the probability. For both factors, threshold and probability, the increase in hypothesis set 1 appears to be exponential, whereas for hypothesis set 2, it seems to be constant. The results presented in both figures demonstrate that evaluating hypothesis set 1 requires a larger sample size than evaluating the hypothesis set 2. This result was anticipated, given that testing of Hypothesis set 1 requires meeting the Bayesian power criterion for both scenarios ($H_0$ and $H_1$); whereas, because the hypotheses in Hypothesis set 2 are complementary to each other, testing the set requires collecting evidence for only one hypothesis [93].

## Discussion and conclusions

This contribution gave an introduction to the design and analysis of cluster randomized behavioral intervention studies when the Bayes factor is used for testing hypotheses. The Bayesian approach has been developed to overcome some of the drawbacks of null hypothesis significance testing. The Bayes factor quantifies support for any hypothesis against another one and Posterior Model Probabilities can be used to select the hypothesis that gets the most support from the data among a set of competing hypotheses. As such, it overcomes the drawback of the *p*-value, which can only quantify support against the null hypothesis but cannot quantify support for the null hypothesis or any other hypothesis. In our empirical example, the *p*-value for the effect of treatment was 0.00639, which is a large degree of support against the null hypothesis. In the Bayesian framework, we compared the null hypothesis to the hypothesis that states that the intervention is doing better than the control. We found a Bayes factor of 63.87 in favor of the latter hypothesis. In other words, there is over 60 times more support in the data for the latter hypothesis that the intervention is doing better than the control than for the hypothesis stating that the two are doing equally well. Such information is very useful but cannot be obtained from null hypothesis testing.

The use of the Bayesian approach to hypothesis testing has broadened and increased over the last few decades. Hence it is important that methodology for a priori sample size determination becomes available. We use a criterion for sample size determination for the comparison of two competing hypotheses, where the sample size should be so large that for both of these hypotheses, a user-selected lower bound of the Bayes factor is exceeded with sufficient probability. We provide an R function to determine the sample size. Our results show that, as in NHST, a larger sample size is needed if the effect size decreases and/or the ICC increases. Furthermore, as is obvious, larger sample sizes are needed for larger values of $BF_{thres}$ and/or η.

It should be noted that Bayesian sequential designs can be used as an alternative to a priori sample size determination [95–98]. With this approach subjects and (in the case of a cluster randomized trial) clusters can be added to the study until the Bayes factor exceeds a user-selected lower bound. It is not necessary to correct for adding subjects, as is the case for traditional group sequential designs [99,100]. However, a (Bayesian) sequential is not always feasible in practice, for instance when a funding agency or ethical committee requires an a priori sample size determination. In longitudinal

trials with a large duration, it may also not be feasible to implement a sequential design. For that reason, it is important methodology for a priori sample size determination is further developed. This will be done for cluster randomized trials and longitudinal intervention studies by the first two authors in a four-year project [93,101] The methodology in this paper will be extended to more complex study designs, such as trials with attrition, multiple outcome variables, studies that compare more than two treatment conditions, and multiperiod cluster randomized trials, such as cross-over and stepped wedge trials.

The reader must be aware of the time and computational demands inherent to the implementation of the methodology presented. As this methodology is based on the simulation of hierarchical data and the fitting of multilevel models, it can be time-consuming. For example, the a priori sample size determination, which was presented using the same values as the sample size determination in the motivating example, took 16.7 minutes to complete. Moreover, on average the simulation study spent 6.14 minutes in each scenario for hypothesis set 1 and 2.55 minutes for hypothesis set 2. If the number of clusters is fixed to be large a priori, then simulating the data and fitting the model is time consuming, meaning that finding the required cluster size is computationally demanding. Similarly, finding the required number of clusters may be computationally demanding as well if the a priori cluster size is large. This in particular occurs when the researcher uses hypothesis set 1. In terms of computational resources necessary for the implementation of the methodology, it should be noted that the function already incorporates the *batch* argument to control the amount of resources used. This is especially useful in computers with limited RAM. In addition, the function employs a binary search algorithm to reduce the amount of iterations. It is anticipated that, as computer technology continues to evolve, the constraints imposed by time and computational cost will be overcome.

Another aspect that researchers must consider when determining the sample size is the specification of the necessary values that play a role in the required sample size. For cluster randomized trials, the value of intraclass correlation coefficient is particularly difficult to guess. Researchers may use values found in literature on the phenomenon of study, or values suggested by experts. One example of the resources available for an educated guess is a summary of papers reporting estimates of ICCs across various scientific fields [88,102]. Another strategy to address the uncertainty surrounding the ICC is to carry out a sensitivity analysis. This analysis consists of using a range of plausible ICCs to determine the sample size, the researcher then may opt to use the largest sample size. This strategy can be extrapolated to other model parameters, such as effect size and the fixed sample size.

We encourage the reader to go over our markdown file and replicate the analyses. We hope the reader will consider the Bayesian approach to hypothesis testing an interesting alternative to null hypothesis significance testing and may consider using it in their own research.

## Supporting information

**S1 File. Example calculation of the Bayes factor.**
(DOCX)

**S2 Fig.** Visualization of fit and complexity for both $H_0$: $\beta = 0$ and $H_1$: $\beta > 0$.
(TIF)

**S3 File. Design and analysis of behavioral intervention studies: A Bayesian approach.**
(HTML)

## Author contributions

**Conceptualization:** Mirjam Moerbeek.

**Formal analysis:** Mirjam Moerbeek, Camila Natalia Barragan Ibañez.

**Methodology:** Camila Natalia Barragan Ibañez.

**Software:** Camila Natalia Barragan Ibañez, Mirjam Moerbeek.

**Writing – original draft:** Camila Natalia Barragan Ibañez, Ulrich Lösener, Nnamdi Moeteke, Mirjam Moerbeek.

**Writing – review & editing:** Camila Natalia Barragan Ibañez, Ulrich Lösener, Nnamdi Moeteke, Mirjam Moerbeek.

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
