## [Decision Letter · Decision Letter 0]

7 Oct 2025

Dear Dr. Barragan Ibañez,

Thank you for submitting your manuscript to PLOS ONE. After careful consideration, we feel that it has merit but does not fully meet PLOS ONE’s publication criteria as it currently stands. Therefore, we invite you to submit a revised version of the manuscript that addresses the points raised during the review process.

The reviewers have requested some clarifications about how some of calculations performed and presented, as well as some methodological details needing more exploration. The reviewers also request a more balanced assessment of the utility of frequentist methods alongside Bayesian analyses.

We look forward to receiving your revised manuscript.

Kind regards,

Christopher Kirk, PhD

Academic Editor

PLOS ONE

Journal Requirements:

3. Please note that PLOS One has specific guidelines on code sharing for submissions in which author-generated code underpins the findings in the manuscript. In these cases, we expect all author-generated code to be made available without restrictions upon publication of the work. Please review our guidelines at https://journals.plos.org/plosone/s/materials-and-software-sharing#loc-sharing-code and ensure that your code is shared in a way that follows best practice and facilitates reproducibility and reuse.

6. Thank you for uploading your study's underlying data set. Unfortunately, the repository you have noted in your Data Availability statement does not qualify as an acceptable data repository according to PLOS's standards.

7. Please include a caption for figure 5.

8. Please ensure that you refer to Figure 5 in your text as, if accepted, production will need this reference to link the reader to the figure.

Reviewers' comments:

Reviewer's Responses to Questions

**Comments to the Author**

1. Is the manuscript technically sound, and do the data support the conclusions?

Reviewer #1: Yes

Reviewer #2: Yes

2. Has the statistical analysis been performed appropriately and rigorously?

Reviewer #1: Yes

Reviewer #2: Yes

3. Have the authors made all data underlying the findings in their manuscript fully available?

Reviewer #1: Yes

Reviewer #2: No

4. Is the manuscript presented in an intelligible fashion and written in standard English?

Reviewer #1: Yes

Reviewer #2: Yes

Reviewer #1: I enjoyed reading your paper and I think it makes a valuable contribution to the literature on Bayesian methods in intervention research. The tutorial approach effectively demonstrates practical implementation of Bayes factors for hypothesis testing in cluster randomised trials, addressing an important methodological gap. Nonetheless, several issues require attention before I can recommend publication.

Critical technical correction required

The most important issue is the contradiction regarding fractional prior specification. You state that b = 1/Neff in the theoretical section but claim b = 1 by default in the implementation section. Since fractional Bayes factors require 0 < b < 1 by construction, this needs immediate correction. Please clarify how Neff is computed for your mixed model and consider providing a brief sensitivity analysis showing stability of Bayes factor conclusions across a small grid of b values.

Methodological clarification needed

Your analysis uses change scores with cluster-level random intercepts and covariate adjustment. Please briefly discuss why this approach was chosen over alternatives (ANCOVA-type analysis or longitudinal mixed models with time × group interactions) that are often preferable for precision in two-time-point CRTs. Additionally, specify which degrees of freedom adjustment method was used and explicitly define which variance component was used for Cohen's d standardisation in your clustered, covariate-adjusted setting.

Since your outcome is a sum of eight three-category items, please acknowledge the approximate-continuity assumption underlying the linear mixed model and note that ordinal alternatives exist when scale properties are uncertain.

Sample size methodology

Your sample-size planning assumes equal cluster sizes and fixes planning at the minimum observed cluster size (4) rather than the empirical average (nearly 12). Please justify this choice and comment briefly on how unequal cluster sizes affect power. The assumption of equal cluster sizes in simulations should be stated explicitly, with comment on the likely impact of non-zero coefficients of variation in cluster size.

Presentation improvements

Ensure consistent reporting of Bayes factor magnitudes throughout the text. The term "fail-safe hypotheses" is non-standard and potentially confusing; use "complement" for Hc and "unconstrained" for Hu while explaining that including Hc ensures coverage of the entire parameter space. Your criticism of null hypothesis significance testing, while valid, occasionally becomes one-sided. A more balanced discussion acknowledging scenarios where traditional methods might be preferable would strengthen credibility.

Reference established, though still heuristic, guidelines for BF interpretation (e.g., Jeffreys' scales or others). This will give readers a concrete starting point, even while you correctly maintain that these scales are interpretive aids, not rigid rules.

Implementation guidance

The paper would benefit from expanded discussion of practical implementation challenges, particularly computational requirements for simulation-based sample size determination. Consider adding guidance on common pitfalls for practitioners attempting to implement these methods.

Minor technical notes

The relationship between different Bayes factor variants could be better explained. The ICC values used for planning (0.054) versus observed (0.042) should be clearly distinguished throughout. Table 1 caption formatting needs correction for footnote markers.

Despite these issues, your manuscript provides clear exposition of Bayes factors via fit and complexity, demonstrates practical utility through a well-executed worked example, and extends the tutorial with valuable simulation-based sample-size determination methodology. With the corrections outlined above, particularly addressing the fractional prior specification and adding the requested methodological clarifications, this will serve as an excellent resource for behavioral and health researchers considering Bayesian approaches to intervention studies.

Finally, your criticism of null hypothesis significance testing, while valid, occasionally becomes one-sided. Acknowledging that there are scenarios where traditional methods might be preferable is a mature and clarifying recommendation that would strengthen your paper's credibility and provide a more balanced discussion for your audience.

Reviewer #2: This is a tutorial on applying Bayesian hypothesis testing, specifically using the Bayes factor, to behavioral intervention studies, with a focus on cluster randomized trials (CRTs). The paper is well-structured, clearly written, and addresses a critical need in the research community by providing a practical alternative to null hypothesis significance testing (NHST).

Please find my comments below:

1)The paper mentions that the prior is calibrated using the effective sample size (N_eff) but does not provide an explicit formula for how N_eff is calculated in the context of a CRT. Please include the formula relating N_eff to the number of clusters, cluster size, and ICC.

2) Equation 5: Equation 5 currently indicates that both conditions must be satisfied simultaneously. However, the subsequent explanation states that for Hypothesis Set 2, either condition alone is sufficient to support the hypothesis. The equation as written does not capture this distinction. The authors should modify Equation 5 to explicitly represent separate criteria for the two hypothesis sets, or add an explanatory note immediately after the equation clarifying how the rule differs between Hypothesis Sets 1 and 2.

3) Line 484: Clarify whether 0.603 is a typographical error for 0.590, or if it was calculated differently. If different, explain which calculation method produced 0.603 and why it differs from Table 1.

4) Figure 4: Figure 4 shows sample size remains constant as ICC increases for hypothesis set 2. This contradicts established CRT theory and requires explanation.

**Do you want your identity to be public for this peer review?** For information about this choice, including consent withdrawal, please see our Privacy Policy

Reviewer #1: No

Reviewer #2: No

---

## [Author Response · Author response to Decision Letter 1]

13 Nov 2025

Response to reviewers

Dear reviewers,

Thank you for the opportunity to submit a revised version of the manuscript “Design and Analysis of Behavioral Intervention Studies: a Bayesian Approach” for publication in the journal PLOS ONE. We appreciate the time and effort taken to review this manuscript. We have made modifications based on the feedback provided by the reviewers, those changes improved the manuscript’s clarity and quality. We provide a file reflecting the changes and another file with a clean version to facilitate the reading. Please note that the page and line numbers correspond to the file reflecting the changes.

All authors have read and approved the revised version of the manuscript. We hope that the revised manuscript is suitable for publication.

Sincerely,

Authors

Reviewer 1

I enjoyed reading your paper and I think it makes a valuable contribution to the literature on Bayesian methods in intervention research. The tutorial approach effectively demonstrates practical implementation of Bayes factors for hypothesis testing in cluster randomised trials, addressing an important methodological gap. Nonetheless, several issues require attention before I can recommend publication.

Authors’ response:

We would like to thank you for affirming the contribution of our manuscript to the existing literature. It pleases us to hear that our tutorial seems to succeed in demonstrating Bayesian hypothesis evaluation and sample size determination in cluster randomised trials. We are very grateful for your insightful feedback on the first version of the manuscript, which prompted us to substantially improve the quality of the manuscript in many areas. In the following, we aim to give you a comprehensive overview of the specific modifications we made to the manuscript in response to your review. We hope that you consider the quality of the paper in its revised form to be sufficient for publication.

Critical technical correction required

1. The most important issue is the contradiction regarding fractional prior specification. You state that b = 1/Neff in the theoretical section but claim b = 1 by default in the implementation section. Since fractional Bayes factors require 0 < b < 1 by construction, this needs immediate correction. Please clarify how Neff is computed for your mixed model and consider providing a brief sensitivity analysis showing stability of Bayes factor conclusions across a small grid of b values.

Authors’ response:

We would like to thank you for pointing towards this error in the implementation section. As you state correctly, the b fraction is by definition larger than zero and smaller than one. This was a typographical mistake, where b = 1 should have been b = 1/Neff. We corrected this passage accordingly (p. 15, line 354).

The formula for Neff was indeed not explicitly provided in the earlier version of the manuscript. Considering the fact that Neff is an essential part of the Bayes Factor calculation, this should absolutely be included in the text. We thank you for drawing our attention to this and have added the formula (along with a reference) to the text (p. 15, line 357).

As for the sensitivity of the Bayes Factor to b, we believe that running an extensive sensitivity analysis would be beyond the scope of this paper. However, we did mention in the text where we refer the interested reader to a published paper of some of the authors where a sensitivity analysis was carried out. We further refer to an existing ShinyApp, in which this sensitivity is visualised in a straightforward way (p. 22, lines 526 to 530).

Methodological clarification needed

2. Your analysis uses change scores with cluster-level random intercepts and covariate adjustment. Please briefly discuss why this approach was chosen over alternatives (ANCOVA-type analysis or longitudinal mixed models with time × group interactions) that are often preferable for precision in two-time-point CRTs. Additionally, specify which degrees of freedom adjustment method was used and explicitly define which variance component was used for Cohen's d standardisation in your clustered, covariate-adjusted setting.

Since your outcome is a sum of eight three-category items, please acknowledge the approximate-continuity assumption underlying the linear mixed model and note that ordinal alternatives exist when scale properties are uncertain.

Authors’ response:

Thank you for noticing this gap. The reason we use multilevel regression is simply in order to preserve comparability with the original study by Moeteke et al. (2024), where the same model is employed. This serves to accurately calculate the Bayesian power based on the sample size in the original study and compare the results of the two sample size determination procedures. However, we acknowledge the fact that this may not have been clear from the text and may leave some readers confused. Therefore, we added an explanation of this rationale in the “Results” section (p. 9, lines 212 to 214).

The degrees of freedom are adjusted according to Satterthwaite’s method as implemented by default in the lmerTest R package. Thank you for pointing out the fact that this was not mentioned previously in the manuscript. We added a mention of this in the note of Table 1 (p. 9).

The standardisation of effect sizes (Cohen’s d) was done according to Hedges and Hedberg (2007) and Hedges (2016), where d=(m1-m2)/sT, where sT2 is the total variance (between cluster + within cluster). We have added a clarification of the fact that this procedure was used to calculate the effect size, along with the relevant references (p. 9, lines 222 to 223).

Thank you for pointing out that the underlying assumption of approximate continuity of the outcome variable should be made more explicit. This assumption indeed does not go without saying, which is why we added a mention of this, along with a reference to an alternative model in case this assumption is not tenable (p. 8, lines 192 to 194).

Sample size methodology

3. Your sample-size planning assumes equal cluster sizes and fixes planning at the minimum observed cluster size (4) rather than the empirical average (nearly 12). Please justify this choice and comment briefly on how unequal cluster sizes affect power. The assumption of equal cluster sizes in simulations should be stated explicitly, with comment on the likely impact of non-zero coefficients of variation in cluster size.

Authors’ response:

Thank you for your observation. At first we decided to use 4 as the cluster size since this is the “worst scenario”, however we agree that using the empirical average is a better option. Therefore, we have carried out the sample size determination again with 12 as fixed cluster size. We have made the change in p. 23, lines 546, 548 and 555. The change is also reflected on Figure 4. The tendencies described in the results were expanded to adapt it to the changes made (see p. 23, lines 548 to 551). Moreover, on pages 20 and 21 lines 487 to 492, there is further explanation about the impacts of assuming equal number of clusters and cluster sizes.

Presentation improvements

4. Ensure consistent reporting of Bayes factor magnitudes throughout the text. The term "fail-safe hypotheses" is non-standard and potentially confusing; use "complement" for Hc and "unconstrained" for Hu while explaining that including Hc ensures coverage of the entire parameter space. Your criticism of null hypothesis significance testing, while valid, occasionally becomes one-sided. A more balanced discussion acknowledging scenarios where traditional methods might be preferable would strengthen credibility.

Authors’ response:

Thank you for pointing out these aspects on the sections “Shortcomings of null hypothesis significance testing” and “Bayesian hypothesis evaluation”. We agree that some of the wording may have been confusing to the reader and modified the text according to your suggestions. We further agree that a more balanced discussion of the two approaches to inference is appropriate. Therefore, we added a subsection “Summary” (p. 12 and 13, lines 298 to 307), in which we clarify that the criticism of p-values in the literature pertains to their use, not their inherent architecture. Also, we highlight some situations in which p-values may be the superior choice compared to Bayesian inference. We furthermore extended our references to existing literature by including a number of articles that relativize the far-spread criticism of p-values.

5. Reference established, though still heuristic, guidelines for BF interpretation (e.g., Jeffreys' scales or others). This will give readers a concrete starting point, even while you correctly maintain that these scales are interpretive aids, not rigid rules.

Authors’ response:

Thank you for raising this point. We are indeed reluctant to provide threshold values that may be misused as cut-offs in the future, leading to problems akin to the ones we see in p-values (Hoijtink et al., 2019). However, we also agree with your point of giving the reader a starting point for interpreting the magnitude of a Bayes Factor. This is why we added a brief description of the most relevant thresholds in Jeffreys (1961), Kass and Raftery (1995), and Lee and Wagenmakers (2014) along with our recommendation to not interpret these too rigidly (p. 13 and 14, lines 323 to 329).

Implementation guidance

6. The paper would benefit from expanded discussion of practical implementation challenges, particularly computational requirements for simulation-based sample size determination. Consider adding guidance on common pitfalls for practitioners attempting to implement these methods.

Authors’ response:

Thank you for this suggestion. We agree that the paper would benefit from a discussion of the potential implementation challenges. Therefore, an explanation of the main limitations that have been identified has been included; namely, the time and computational costs. A mention of the measures used in the algorithm in order to improve the control of computational costs is also included. Another point that is discussed is the uncertainty of the ICC. Researchers must use an educated guess of ICC using previous research in the phenomenon of study or suggestions by experts. We provide a reference to a summary of papers reporting the estimated ICC across various scientific fields. In addition, an explanation is provided for the application of a sensitivity analysis to address uncertainty. These changes are reflected on pages 27 and 28, lines from 649 to 674.

Minor technical notes

7. The relationship between different Bayes factor variants could be better explained. The ICC values used for planning (0.054) versus observed (0.042) should be clearly distinguished throughout. Table 1 caption formatting needs correction for footnote markers.

Authors’ response:

Thank you for drawing our attention to these three minor issues. While we believe that an extensive overview of variants of Bayes Factor computation is beyond the scope of this paper, we do agree that some context on the motivation of using the approximate adjusted fractional Bayes Factor is appropriate. We therefore added a brief comment (p. 14, lines 332 to 336) on the three most notable advantages of this way of calculating the Bayes Factor over others: Its straightforwardness, the default fractional prior which removes the potentially burdensome task of specifying a prior by hand, and its fast computation due to the absence of a Markov Chain Monte Carlo sampling mechanism. Note that the latter is of special importance in simulation settings as is the case in the presented method.

We have added the terms “observed” and “planning” to make the distinction when this was not clear with the context. The changes are reflected in line 451 on page 19, and line 532 on page 22.

Finally, we corrected the Table 1 caption formatting on page 9.

8. Despite these issues, your manuscript provides clear exposition of Bayes factors via fit and complexity, demonstrates practical utility through a well-executed worked example, and extends the tutorial with valuable simulation-based sample-size determination methodology. With the corrections outlined above, particularly addressing the fractional prior specification and adding the requested methodological clarifications, this will serve as an excellent resource for behavioral and health researchers considering Bayesian approaches to intervention studies.

Finally, your criticism of null hypothesis significance testing, while valid, occasionally becomes one-sided. Acknowledging that there are scenarios where traditional methods might be preferable is a mature and clarifying recommendation that would strengthen your paper's credibility and provide a more balanced discussion for your audience.

Authors’ response:

We want to thank you again for your insightful feedback and your attention to detail. Your perspective on the value of this contribution is greatly appreciated and we are happy to hear that you deem this method and the accompanying tutorial a valuable resource for applied researchers. We sincerely hope that the modifications outlined above pertain to your concerns about some methodological and textual aspects of the manuscript.

We fully acknowledge your criticism of the section on drawbacks of null hypothesis significance testing (NHST) and agree that a more balanced discussion is appropriate. Hoping to achieve this, we added a subsection clarifying that the aforementioned drawbacks pertain to the use of p-values, rather than their inherent structure (p. 12 lines 298 to 307). We also outline situations in which p-values may be preferable over Bayes Factors as inferential tools and provide the reader with some references to literature relativising some of the most prominent criticisms of NHST.

We appreciate you taking the time and making the effort to help us improve upon this research project. We believe that your input has greatly strengthened this manuscript.

Reviewer 2

This is a tutorial on applying Bayesian hypothesis testing, specifically using the Bayes factor, to behavioral intervention studies, with a focus on cluster randomized trials (CRTs). The paper is well-structured, clearly written, and addresses a critical need in the research community by providing a practical alternative to null hypothesis significance testing (NHST).

Please find my comments below:

1. The paper mentions that the prior is calibrated using the effective sample size (N_eff) but does not provide an explicit formula for how N_eff is calculated in the context of a CRT. Please include the formula relating N_eff to the number of clusters, cluster size, and ICC.

Authors’ response:

Thank you for drawing our attention to this omission. We agree that the formula for Neff should be included in the text and have added it accordingly along with a reference to a textbook with further elaboration on this formula (p. 15, line 357).

2. Equation 5: Equation 5 currently indicates that both conditions must be satisfied simultaneously. However, the subsequent explanation states that for Hypothesis Set 2, either condition alone is sufficient to support the hypothesis. The equation as written does not capture this distinction. The authors should modify Equation 5 to explicitly represent separate criteria for the two hypothesis sets, or add an explanatory note immediately after the equation clarifying how the rule differs between Hypothesis Sets 1 and 2.

Authors’ response:

Thank you for pointing out this discrepancy. We agree that Equation 5 does not represent the criteria for the two hypothesis sets. We have included, on page 18 lines from 427 to 429, a new equation defining the power criterion for Hypothesis set 1. Beside the inclusion of another Equation, we have restructured the paragraphs to make the text more coherent and highlight the differences between the two sets.

3. Line 484: Clarify whether 0.603 is a typographical error for 0.590, or if it was calculated differently. If different, explain which calculation method produced 0.603 and why it differs from Table 1.

Authors’ response:

Thank you for pointing out the discrepancy. This is indeed a typographical error that has been

---

## [Decision Letter · Decision Letter 1]

15 Dec 2025

Dear Dr. Barragan Ibañez,

We look forward to receiving your revised manuscript.

Kind regards,

Christopher Kirk, PhD

Academic Editor

PLOS One

Journal Requirements:

Reviewers' comments:

Reviewer's Responses to Questions

**Comments to the Author**

Reviewer #1: All comments have been addressed

Reviewer #2: (No Response)

2. Is the manuscript technically sound, and do the data support the conclusions?

Reviewer #1: Yes

Reviewer #2: Yes

3. Has the statistical analysis been performed appropriately and rigorously?

Reviewer #1: Yes

Reviewer #2: Yes

4. Have the authors made all data underlying the findings in their manuscript fully available?

Reviewer #1: Yes

Reviewer #2: (No Response)

5. Is the manuscript presented in an intelligible fashion and written in standard English?

Reviewer #1: Yes

Reviewer #2: (No Response)

Reviewer #1: Thank you for the revisions, you have adressed my questions and concerns. Pointing to the interactive Shiny app provides readers with a useful pedogogical tool, but it took me some time to find the shiny app. I would explicitly include a link for readers to easily follow: https://utrecht-university.shinyapps.io/BayesSamplSizeDet-CRT/

Reviewer #2: The authors have revised the manuscript adequately. There is only one minor comment.

While the authors explained the line in Figure 4 is flat because of a floor effect in their simulation algorithm, they did not explicitly include this "minimum boundary" explanation in the main text. Please refine the text stating that the minimum sample size investigated (8 clusters) was sufficient to meet the power criterion across the tested ICC range. Additionally, please include a brief note in the Figure 4 caption and the main text clarifying that this flat line represents the lower bound of the simulation grid, which effectively masks the minor influence of the ICC at this specific effect size.

**Do you want your identity to be public for this peer review?** For information about this choice, including consent withdrawal, please see our Privacy Policy

Reviewer #1: No

Reviewer #2: No

---

## [Author Response · Author response to Decision Letter 2]

6 Jan 2026

Dear reviewers,

We appreciate the time and effort taken to review this manuscript. We have made minor modifications following the feedback provided. We provide a file reflecting the changes and another file with a clean version to facilitate the reading. Please note that the page and line numbers correspond to the file reflecting the changes.

Reviewer 1

Thank you for the revisions, you have addressed my questions and concerns. Pointing to the interactive Shiny app provides readers with a useful pedagogical tool, but it took me some time to find the shiny app. I would explicitly include a link for readers to easily follow: https://utrecht-university.shinyapps.io/BayesSamplSizeDet-CRT/

Authors’ response: We would like to thank you for recognising the added value of the Shiny app. We have included in the main text on page 22, line 511, the link to the Shiny app.

Reviewer 2

The authors have revised the manuscript adequately. There is only one minor comment. While the authors explained the line in Figure 4 is flat because of a floor effect in their simulation algorithm, they did not explicitly include this "minimum boundary" explanation in the main text. Please refine the text stating that the minimum sample size investigated (8 clusters) was sufficient to meet the power criterion across the tested ICC range. Additionally, please include a brief note in the Figure 4 caption and the main text clarifying that this flat line represents the lower bound of the simulation grid, which effectively masks the minor influence of the ICC at this specific effect size.

Authors’ response: Thank you for pointing the absence of information that would facilitate the interpretation of the results. We have included on page 24, lines 561 and 562, a sentence explaining that the constant line reflects the minimum sample size assumed in the algorithm. Furthermore, on the same page, lines 564 to 565 and 569 to 570, we have included the minimum sample size chosen in the algorithm. In addition, following a revision of Figure 4, we adjusted the scale of the upper left plot to start at zero, thus ensuring a common start of the y-axis across all plots.

---

## [Decision Letter · Decision Letter 2]

19 Jan 2026

Design and Analysis of Behavioral Intervention Studies: a Bayesian Approach

PONE-D-25-33347R2

Dear Dr. Barragan Ibañez,

We’re pleased to inform you that your manuscript has been judged scientifically suitable for publication and will be formally accepted for publication once it meets all outstanding technical requirements.

Kind regards and congratulations,

Christopher Kirk, PhD

Academic Editor

PLOS One

Additional Editor Comments (optional):

Reviewers' comments:

Reviewer's Responses to Questions

**Comments to the Author**

Reviewer #2: All comments have been addressed

2. Is the manuscript technically sound, and do the data support the conclusions?

Reviewer #2: (No Response)

3. Has the statistical analysis been performed appropriately and rigorously?

Reviewer #2: (No Response)

4. Have the authors made all data underlying the findings in their manuscript fully available?

Reviewer #2: (No Response)

5. Is the manuscript presented in an intelligible fashion and written in standard English?

Reviewer #2: (No Response)

Reviewer #2: (No Response)

**Do you want your identity to be public for this peer review?** For information about this choice, including consent withdrawal, please see our Privacy Policy

Reviewer #2: No

---

## [Editor Report · Acceptance letter]

PONE-D-25-33347R2

PLOS One

Dear Dr. Barragan Ibañez,

I'm pleased to inform you that your manuscript has been deemed suitable for publication in PLOS One. Congratulations! Your manuscript is now being handed over to our production team.

Kind regards,

on behalf of

Dr. Christopher Kirk

Academic Editor

PLOS One